# Towards a Pentecostal Homiletic: A Re-Enchanted Methodology

Taylor S. O. Drake

Department of Christian Studies, Southwestern Christian University, Bethany, OK 73008, USA; taylor.drake@swcu.edu

**Abstract:** As Pentecostalism continues to develop distinct processes, practices, and pedagogies as a unique worshipping community within the Church, little attention has been given to the production of a Pentecostal homiletic, or, at the very least, whether there is a need for one. Since Pentecostal hermeneutics has continued to evolve and solidify over the past century, it naturally follows that the next question to answer would be this: how do Pentecostals produce their sermons? This paper will address the philosophies of the Church; how worldview/hermeneutics/didoiesis build upon each other within the worshipping community; how Pentecostals view the church service, with specific attention to the sermon; and what Pentecostal homiletics would look like practically. Additionally, this paper argues that not only do Pentecostals provide renewal and re-enchanted views of scripture: but they also establish a unique 11-step homiletic, building upon the works of Karl Barth, James K.A. Smith, Chris E.W. Green, and Cheryl Bridges Johns.

**Keywords:** Pentecostal; homiletics; philosophy; hermeneutics; enchantment; worldview; theology; sermon

## 1. Philosophies of the Church

### 1.1. General Church Philosophies

There is an adage that suggests everyone is a philosopher. If philosophy is understood to mean rationally thinking about the general nature of the world (metaphysics), justification of belief (epistemology), and conduct of life (ethics) (Honderich 1995), then there is a good argument to be made that all men are indeed philosophers, though the quality of their philosophy may be subject to questioning. With the above definition, it should be understood that there are several different schools of thought regarding metaphysics, epistemology, and ethics, and much of the conversation is based on either assumptions or presuppositions.

The Church is no different.[1] Whereas theology may be lauded as the Church's unique contribution to philosophy, theology and philosophy overlap so effortlessly that it is often difficult to distinguish between them. In his groundbreaking work, *Tongues of Fire: A Systematic Theology of the Christian Faith*, Dr. Frank D. Macchia provides a frighteningly simple definition for theology: God-talk (Macchia 2023). The brevity of this definition should not dissuade readers from taking it seriously. God-talk is a multifaceted exercise that includes both God and man, in concert with and, oftentimes, against each other. To speak about God, whether spoken by a believer or unbeliever, is always God-talk. In light of the above adage, perhaps RC Sproul's idea that everyone is a theologian is not so foreign.[2] Like philosophy, the quality of theology is what is being brought into question.

A simplified definition of philosophy is perhaps obtainable in light of Macchia's God-talk. This author suggests that if everyone engages in theology by speaking about God, then everyone also engages in self-talk; that is to say, philosophy is self-talk. While philosophy addresses metaphysics, epistemology, and ethics, it does so with a person's self at its center; it is the self's understanding, experiences, and ideas that build and inhabit philosophy. Broadening the definition is perhaps beneficial from the self to humanity. Therefore, philosophy is humanity's interpretation of its own self, particularly within/of

the world, which—though this is a discussion for another time—suffices to say that theology and philosophy are not opposed to each other by definition.

So, why all this talk about theology and philosophy? To put it simply: the Church has philosophies. This is where the line distinguishing theology from philosophy and vice versa begins to proverbially "blur". The Christian faith proposes that humans are holistic beings composed of interconnecting parts: body and soul.[3] Within these parts are other facets that add to humanity's complex ontology: emotions, passions, experiences, memories, perceptions, etc. However, the Church has provided distinct perspectives and responses to the conversation of metaphysics, epistemology, and ethics, often against popular or "modern" opinions on the matter. Why is this the case? Is it not because the Church begins her philosophies with God, who is confessed to be perfectly revealed in Jesus Christ, encountered in the Christian scriptures through the illumination of the Holy Spirit? The Church's philosophies begin upon the presupposition that reality is real, created, and actual, and that God created humanity to not only know God but to also know his meaning, value, and purpose within creation in relationship to the creator.[4] With this in mind, it follows that the Church philosophizes differently to "the World".

It seems right to this author to recognize that the Church has its own—or at the very least, a redeemed usage/practice regarding—philosophies. These philosophies include but are not limited to worldview (a semiotic system of narrative signs that has a significant influence on the fundamental human activities of reasoning, interpreting, and knowing (Naugle 2002, p. 253)), hermeneutics (the art and science of interpretation (Virkler 2023, p. 2)), and didoiesis (the art of making doctrines to teach[5]). Special attention should be given to didoiesis, as it is a unique blend of the spheres of pedagogy and catechism. In other words, worshipping communities are not only "being made by" and "making/have made" doctrines: they have also established the teachability and teaching methods for those doctrines. Rightly understood, these three specific philosophical activities both flow from and into each other, interdependently influencing one another, within the worshipping community, both church and academy. However, as stated above, these are philosophical activities that are not performed in a vacuum; they are "redeemed" or even "sanctified" philosophies, restored to their original purposes by the active presence of the Holy Spirit within the Church. However, these philosophies present an opportunity for the distinct worshipping communities within the Church to contribute their own unique "take" on these philosophies. This in turn requires those worshipping communities to define their "takes" clearly within the Church, establishing a tension of accountability wherein the other worshipping communities within the Church provide "checks and balances" concerning the dogma of the faith whilst respecting the worshipping community's doctrine.[6]

### 1.2. General Pentecostal Philosophy

Within the Church are many rooms, and perhaps the loudest of the rooms, currently, is the Pentecostal church.[7] The past two decades have seen the production of numerous influential texts from Amos Yong, Kenneth Archer, and Cheryl Bridges Johns, to name a few. However, the start of the Pentecostal philosophical movement can be directly traced back to James K.A. Smith's manifesto *Thinking in Tongues*, through which Smith almost singlehandedly reinvigorated philosophy within the Pentecostal community. His proposal is simple: Pentecostalism, as a spirituality, is best seen as a worldview and therefore contains five key aspects (Smith 2010):

1. Radical openness to God: God through his Spirit is seen as actively involved in his creation, bringing about things that are different or new, echoing Peter's message in Acts 2.
2. "Enchanted" theology of creation and culture: creation has metaphysical components and aspects (the Spirit and other spirits) that may manifest through various means and expressions.
3. Non-dualistic affirmation of embodiment and materiality: the anthropological ontology of an individual's being acknowledges divine healing as a potential reality.

4.  Affective, narrative epistemology: experience (e.g., the imagery of the "heart") is valued as a means of "knowing things"; therefore, rationality is favored over and above rationalism (Smith 2010, pp. 52–58).[8]

5.  Eschatological orientation to mission and justice: the marginalized are not only considered but valued, so that the Pentecostal philosopher can appropriately live out biblical social, as opposed to secular social, justice.

These principles depict a unique view of Pentecostalism within the Church, especially since Pentecostalism is viewed as a renewal movement.[9] In a sense, a Pentecostal worldview would therefore be considered a worldview within a worldview. This is nothing truly exceptional as denominations generally affirm the dogmatic requirements to be considered "Christian" while possessing unique features or expressions of those dogmatics. In other words, Pentecostalism is a Christian worldview within the Church's worldview.

Steven Félix-Jäger and Yoon Shin have added to this notion of a Pentecostal worldview by defining a renewal worldview as a fundamental orientation of the narrated body that implicitly and often subconsciously imagines and understands reality.[10] While this may seem to contradict the traditional Christian worldview[11], the Pentecostal worldview erupts from the pages of scripture, quite literally, producing something again that has been interrupted.[12] With this in mind, a question naturally arises: which is produced first—worldview, hermeneutic, or didoiesis? This author suggests that the question reveals a faulty assumption; namely, that these three components of the worshipping community are not produced simultaneously. Instead, it is argued that these three components build both from and upon each other. Worldview, hermeneutics, and didoiesis work interdependently within and from the worshipping community to address the perspective, process, and pedagogy of the worshipping community. In other words, the worshipping community is identified by their perspective, process, and pedagogy, and the perspective, process, and pedagogy also identify the worshipping community. The worshipping community is both "making" and "being made" by these three aspects.

However, what one discovers is that worldview, hermeneutics, and didoiesis are all respondents to a thing, specifically an event/experience (E).[13] Following the event/experience, worldview (W), hermeneutics (H1), and didoiesis (D) are produced by the worshipping community (WC) to not only make sense of things but to also embody and affirm the event/experience. These are then communicated through the sermon (S) whereby God (G), the worshipping community (WC), and neighbors (N) encounter and experience one another. This progression can be seen in Figure 1 below. H2 will be addressed later.

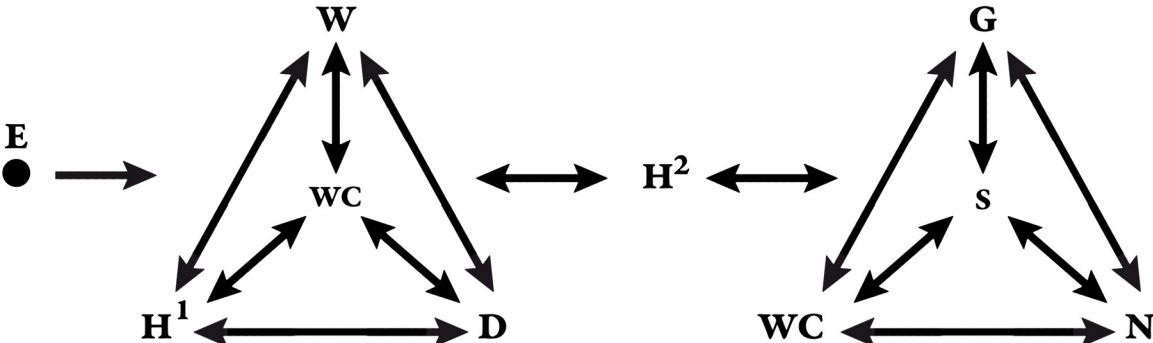

**Figure 1.** Model of Worshipping Community Creation and Production.

For the Church, the Pentecostal movement (primarily the unique events/experiences with God (i.e., manifestations of the Holy Spirit)) could not be adequately explained with the Church's current processes, practices, or pedagogies, and so it demanded a better, more holistic explanation. Pentecostalism then proceeded over decades to produce a renewed worldview (Félix-Jäger and Shin 2023), renewed hermeneutics, and renewed didoiesis

([Land 2010](#)). Here, the question naturally turns toward the "heart" of most Protestant church services: the sermon.

### 1.3. The Sermon

It seems clear that the primary means through which the Church communicates its worldview, hermeneutics, and didoiesis is through the church service.[14] This author recognizes that there is no universal uniformity to how the churches within the Church conduct and order their services.[15] Therefore, the remainder of this paper will focus specifically on how Pentecostals regard the service, specifically the sermon.

Now, regarding this paper's topic, "Towards a Pentecostal Homiletic", the sermon is of utmost importance. Considering the above image, the event of Pentecost (E) had produced distinctly Pentecostal worldviews (W), hermeneutics H1), and didoiesis (D) which are expressed through the sermon (S) where in God (G), the worshipping community (WC), and the neighbors (N) encounter one another. This, however, presents two underlying inquiries that Pentecostalism should address: (1) what happens during the sermon (S), and (2) how is such a sermon composed (H2)?

## 2. The Pentecostal Sermon

### 2.1. Green's Principles

Regarding the first of the two inquiries, Chris E.W. Green outlines a distinctly Pentecostal groundwork towards a theology for preaching in "*Transfiguring Preaching: Salvation, Meditation, and Proclamation* ([Green 2015](#), pp. 69–81)". Here, Green's ideas could be worked out so that preaching is how the worshipping community learns to "speak savingly" from the pastor ([Green 2015](#), pp. 69–70).[16] While preaching is proclaiming the Gospel, "proclaiming the Gospel" entails far more depth than simply retelling the Christ story. This author agrees with Green's six aspects of Pentecostal preaching; but, more importantly, what these six aspects imply happens during the sermon's presentation and reception/rejection.

1. Theotic encounter: "Preaching is a divine-human event that brings God in humanity in the dialogue, partnership, and communion—and in that event, the church is the church ([Green 2015](#), p. 71)". In other words, the sermon is where, through the voice of God in the mouth of the minister, the worshipping community and God meet.

2. Event of/occasion for the Spirit: "preaching... Must occasion a hearing from God, an encounter with God... (preaching) fuses spirit and Word in the sacramental space of the worship event and this fusion creates a zone of revelatory, efficacious grace that causes the sermon to convey transformative power...(depicting) preacher and congregation 'overwhelmed and transformed by the sheer interruptive power of mystery itself ([Green 2015](#), pp. 72–73)'". In other words, through sermons by the ever-presence of the Holy Spirit, the worshiping community's experiences of God's activity in the text are re-enchanted.

3. Overwhelmed, overwhelming askesis: "The deifying efficacy of preaching takes shape of language begins to bend, as the preacher and congregation are disoriented by a text, as the preacher finds a way to allow the coming-undone-ness of her words to be like the breaking of the alabaster jar ([Green 2015](#), p. 75)". In other words, through the sermon, the worshipping community adopts Jacob's posture of both wrestling with, being deformed by, and being put together again by God's words.

4. Protest for/against god: "The preacher and congregation together receive a faithful wound, receiving affliction...Faithful preaching, then, is conceived, born, delivered, and received in fear and trembling as well as a delight and longing, and only so is it an effectual ([Green 2015](#), pp. 75–76)". In other words, through the sermon, the worshipping community acknowledges God's disciplinary activity as well as his grace as both a sanctifying wounding and healing.

5. Holy Spirit and (un)holy spiritedness: "The Spirit cannot be reduced to the preacher's spiritedness...(but) the Spirit's movements cannot be tracked by simply watching what happens in the congregation's response to a sermon ([Green 2015](#), p. 77)". In

other words, through the sermon, the worshipping community and the Spirit are discerning each other, both speaking and listening, waiting and acting, observing and participating.

6.  Kenosis and diakonia: "...Preaching is as a peculiarly burdened, bold but uncertain speaking that can never be hurt but can, by dint of its divine strangeness, make hearing possible...like manna: it sustains us only as it also tests us because it does not fit our tastes (Green 2015, pp. 79–80)". In other words, the sermon is a type of ministry that invites the worshipping community's preferred selves to be "poured out" so they may be filled by the Spirit their true selves.

For Green and the Pentecostal minister, God—through the sermon—both proclaims and invites the participation of the worshipping community to live out the Christian Life before their neighbors in humility, love, and grace. All of Green's principles are diluted to a single idea: the worshipping community encounters the voice of God in the sermon from scripture for their sanctification before their neighbors. The Pentecostal sermon is both convicting and comforting, wounding and healing, disorientation and reorientation.

As a note, Green, like other Pentecostal scholars and theologians, places a special emphasis on principles over and even above methods. Perhaps this is in line with Land's notion that Pentecostalism as spirituality (as opposed to denomination) works better. While principles produce methods, methods are not required for principles. This allows an ambiguity in practice that recognizes culture, context, and community as potential influencers on how principles are practiced; that is, how principles are expressed through the method.

### 2.2. The Worldview of the Pentecostal Sermon

With the above worldview and principles established, it is not hard to see how Green's principles work neatly with Smith's Pentecostal worldview prescriptions. Indeed, they appear to have almost direct correlations with one another:

1.  Radical openness to God *towards* theotic encounter: both emphasize the need for God to sanctify the worshipping community in his way.
2.  "Enchanted" theology of creation and culture *towards* event of/occasion for the Spirit and Holy Spirit and (un)holy spiritedness: all three acknowledge the activities of Spirit and spirits within/without the worshipping community.
3.  Non-dualistic affirmation of embodiment and materiality *towards* overwhelmed, overwhelming askesis: both emphasize the Möbius nature of man as an embodied soul.[17]
4.  Affective, narrative epistemology *towards* protest for/against God: both emphasize that God's activity is primarily relational.
5.  Eschatological orientation to mission and justice *towards* kenosis and diakonia: both emphasize the "loving your neighbor as yourself" and the "of the least of these" aspects of Christian service.

Green and Smith answer their questions so similarly that both philosophy and theology appear to be dancing in tandem, neither stepping on the other's foot while both move gracefully, effortlessly leading and following as needed as when one strengthens the other's weaknesses. From this perspective, both God-talk and self-talk emerge as striving for the same goal: less attempting to explain or defend God (apologetics) and more encouraging of encountering God through what is said and heard, with or without his words. From this synthesis of worldview and sermon principles, homiletics can be considered and realized.

### 3. Hermeneutics vs. Homiletics

It is here that the issue of homiletics (H2) must be considered. Whereas hermeneutics is the art and science of interpretation, homiletics appears to be far more a philosophy of the Church than worldview, hermeneutics, or didoiesis, since homiletics is the art and science of writing sermons and preaching. Is homiletics a philosophy? It is. More specifically, it is the communication of the Church's worldview, hermeneutics, or didoiesis, as represented by H2.

As mentioned above, homiletics is twofold. First, it is the art and science of writing a sermon and second, it is the art and science of preaching. With this definition present, Green's principles towards Pentecostal preaching are homiletic; moreover, they are holist homiletic principles. Much like how worldview, hermeneutics, and didoiesis are interdependently communicating with each other, so too are the writing and the preaching of the sermon. Both preaching and writing "make" and are "being made" by the sermon.

Consider the preacher preparing her sermon. Does she not imagine herself before the worshipping community, performing her sermon (Barth 1991, pp. 71–75)?[18] Does she not consider the faces of the worshipping community, perhaps anticipating their responses or lack thereof? Is she not aware of their strengths and weaknesses, their histories and narratives, their wounds and scars? Does she not know her worshipping community; that is to say, "where they are spiritually?" Of course, she does, since she is able.

It is here that the preacher prepares to speak both from and to the worshipping community. Whereas hermeneutics describes how to interpret a Scripture, homiletics describes how to write about and from scripture within the worshipping community as they encounter God and their neighbor. All of this was occasioned by the Spirit.

It is here that Kenneth J. Archer must be discussed. In his monumental text, *A Pentecostal Hermeneutic*, Archer suggests that whatever the Pentecostal hermeneutic is, it must occur within the interdependent tridactic dialogue between scripture, the Spirit, and the community, resulting in a creative negotiated meaning (Archer 2005, p. 260)". For Archer, while there are definitive aspects of what the Pentecostal should consider while engaging in discerning meaning for the worshipping community, those aspects are largely contained within the worshipping community. This further emphasizes this author's point that hermeneutics, worldview, and didoiesis are activities of the worshipping community. Archer could add that scripture and Spirit play a vital role in that identification process, which this author would affirm. It takes no effort to suggest that the Holy Spirit is around and within the process, as seen in Figure 2 below.

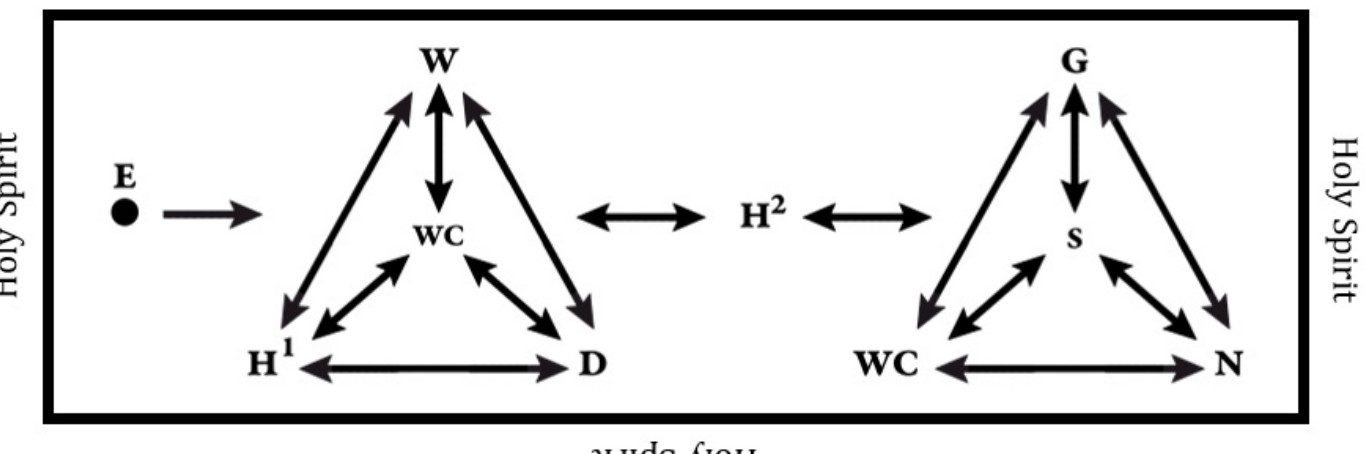

**Figure 2.** Edited Model of Worshipping Community Creation and Production.

It is no suggestion at all for the Pentecostal: she believes that if the Spirit is intimately within the community then the Spirit is intimately within her and her activities! Archer's ideal hermetical process is preserved and built upon in the above illustration. With Smith's worldview, Green's principles, and Archer's hermeneutics in mind, the homilist can consider Barth, Buttrick, and modern homiletics.

## 4. Barth and Other Homiletics Types

### 4.1. Barth's Homiletics

Since Karl Barth stands as the dominant theological mind of the 20th century, special consideration should be given to his work in the field of homiletics. While Barth's own Swiss Reformed affiliation colors his work, not even Pentecostals can escape the Barthian shadow.[19] Green (and Johns below) references Barth in his work, reminding us that Bart considered preaching to be "dancing on the edge of mystery (Green 2015, p. 73)". Indeed, Barth's homiletics provide a definitive starting point for modern homiletics. Here, Barth provides two lists for ministers when composing their sermons (Barth 1991). The first is the Criteria of the Sermon:

1. Revelation (Barth 1991, pp. 47–55), wherein God is both the object and subject of the preaching, is neither provable by intellectual demonstration nor in creating the reality of God. Natural theology (knowledge of God based on observed facts and experience apart from divine *revelation*) has no place within the kerygmatic event.[20]
2. *Church*, wherein the worshipping community "gathers around" this kerygmatic event (Barth 1991, pp. 55–63).
3. *Confession* (*commission*), wherein during the kerygmatic event, *professing the faith* and "*by obedience to obedience*" of the worshipping community happens (Barth 1991, pp. 63–66).
4. *Ministry*, wherein the kerygmatic event happens from within the worshipping community and becomes the *activity* of the worshipping community (Barth 1991, pp. 66–71).
5. *Heralding* (*holiness*), wherein the kerygmatic event includes the proclamation of the kerygmatic event itself within the worshipping community, to be taken outside the "church" by its members (Barth 1991, pp. 71–75).
6. *Scripture*, wherein the kerygmatic event, the exposition of scripture is above all else treated as God's Word revealed in God's words (Barth 1991, pp. 75–81).
7. *Originality*, wherein the kerygmatic event the minister personally identifies with and subjects herself to God's Word in God's word; only then is she "free" to preach.
8. *Congregation*, wherein the kerygmatic event, God's Word of God's word appears contextually to certain people within a certain time and certain place for their "faithing" as members of and within the worshipping community (Barth 1991, pp. 84–85).
9. *Spirituality*, wherein the kerygmatic event happens with humility, soberness, and prayer (Barth 1991, p. 86).

Within these criteria, the Pentecostal should not bristle against these principles. It seems obvious that both Barth and Green are discussing what happens *during* the sermon. Barth, however, does not end here. Whereas the above are indeed criteria, that is *a principle or standard by which something may be judged or decided*, he moves beyond criteria and provides "Actual Preparation of the Sermon (Barth 1991, pp. 91–134)". Here, Barth establishes an order or method, which can be summarized as such:

1. *Preparing sermon preparation*: this includes text selection (what *must* and *must* not be done in the sermon) and properly remembering hermeneutical concerns.
2. *Sermon preparation–receptive*: this includes reading the text and inquiring about the text's contents through the use of additional recourses (commentaries, etc.)
3. *Sermon preparation–spontaneous*: this includes recognizing the theme (*scopus*) of the text (since the Bible is both document and monument in and of the church), answering the question of "how" in the application of the text to faithing in and outside of the worshipping community, whether the message of the text is *optimum* or *pessimum*, actually writing the sermon (totality, introduction, parts, and conclusion), and considerations given to language and presentation.

Is it any wonder that the Pentecostal minister enjoys Barth? Even his methodology is still vague enough to suggest that his method is just an ordering of principles: (1) take the passage seriously as God's word in God's word, (2) give weight to the interpretation for the worshipping community of the passage as God's word in God's word, (3) organize

the study into a presentational form to proclaim God's word in God's word, and then (4) preach God's word as God's word!

The intention of Barth is very clear: whatever occurs during the sermon can only occur within the worshipping community. This event is unique: for Barth, it is the kerygmatic event. It is for the Pentecostal, too. However, the question has not been answered: how does one write the sermon? Barth here is helpful with his three-step method, but his methodology is an option in homiletics.

### 4.2. Buttrick's Homiletics

David Buttrick's *Homiletic: Moves and Structures* begins with perhaps the most frustratingly acute observation about the issue: *"Homiletics is an odd discipline"*(Buttrick 1987, p. xi). While Buttrick's work may seem antiquated (though not as removed from the modern discussion as Barth's), his work demands consideration as he emphasizes more than just the process of sermon writing. His particular attention to the minister's psychology, biases, influences, culture, and other personal and communal aspects are points few other authors bring into consideration within this pedagogical framework. As far as this author can tell, most of these considerations are reserved for hermeneutical activities, rather than homiletical ones. So, for this paper, Buttrick's methods still provide a rudimentary foundation for future homilists to consider.

For Buttrick, *moves*[21] focus on naming (giving words to things) and narration (telling God's story where one finds herself) (Buttrick 1987, pp. 6, 11–13, 20)[22], while *structure* focuses on how and where the *naming* and *narration* are organized to compose the sermon as well as where the sermon is placed within the service (Buttrick 1987, pp. 251–62, 285–317). *Moves* could be seen as the "notes within the chosen key" while the *structure* is "how, when, and where the notes are played" if a musical analogy can be employed. With this understanding, *moves* focus on all the information gathering regarding the text that is being preached upon, whereas *structure* focuses on the order of that information while considering the most appropriate means of communicating that information. Regardless, Buttrick reminds the reader that whatever the *moves* and *structure*, the ultimate goal is for "faith (to) come from sharing (Buttrick 1987, p. xii)".

In his *moves*, Buttrick points the preacher to consider the following:

1. Models of communication
2. Images and metaphors
3. Developing flow within sermons
4. Use of examples and illustrations
5. Narrative theology
6. Style and language
7. Point-of-view

While Buttrick covers more in his five-hundred-page book, the above items serve to illustrate this author's point: *moves* are the *stuff* of sermons.[23] The above items finally move from the "spiritual" aspects of Green and Barth to the more "practical" aspects of homiletics. The Pentecostal preacher needs to be trained not only in "discerning the Spirit" but also in "studying to show oneself approved".

In his *structures*, Buttrick points the preacher to consider the following:

1. What is the authority from preaching?
2. Modes of preaching
3. Where the sermon belongs in the service
4. Forms of preaching (Old and New Testament)
5. Homiletic theories

Like *moves*, this list is not exhaustive but serves to further this author's point: *structures* are deeply concerned with the *presentation* of the *stuff*. These ideas move even further from Green and Barth's "spiritualness" into the even deeper "practical". However, this author is not so convinced Green and Barth would object to Buttrick's considerations. Here, Buttrick

could be seen as the lead-up to "what happens in preaching/the sermon", where Green and Barth's ideals are realized. In other words, the Pentecostal preacher's work alongside the Spirit may very well be found in the intentional discipline of her homiletical skills while discerning both Spirit and Church through and from scripture. Buttrick's work establishes a deeper consideration of the narrative aspects of the sermon within the worshipping community's liturgy. Furthermore, as Buttrick emphasizes, the homilist should take into deep consideration her own biases, presuppositions, influences, and even agendas when engaging in homiletics for her worshipping community.

*4.3. Regarding the (Post)Modern Discussion of Homiletics and Pentecostalism*

In the recent *Homiletics and Hermeneutics: Four Views on Preaching Today*, editors Scott M. Gibson and Matthew D. Kim compile Bryan Chappell (redemptive-historic), Abraham Kuruvilla (Christiconic), Kenneth Langley (Theocentric), and Paul Scott Wilson's (Law-Gospel) essays and responses into a helpfully concise volume for any homilist (Gibson and Kim 2018). However, as recognized above, most texts hold hermeneutics above homiletics, as clearly illustrated by the amount of attention given to homiletics as distinct from hermeneutics (Gibson and Kim 2018).[24] Indeed, only Langley and Wilson offer anything directly concerning homiletical methods.[25]

For Langley, the Homiletical rationale for his theocentric view is (Gibson and Kim 2018, pp. 95–99) as follows:

1. Sermons should exposit the God-breathed text.
2. Sermons should be prepared and delivered in the power of the Holy Spirit.
3. Sermons should model good hermeneutics.
4. Sermons should be weighty.

Langley continues from these principles, flushing out his application rationale as follows (Gibson and Kim 2018, pp. 99–106):

1. Use theocentric vocabulary when interrogating the text.
2. Make God the subject of many of the sermon's sentences.
3. Don't hesitate to exhort.
4. Privilege the text.

It should be obvious that Langley's homiletic points to his hermeneutic. This is not an issue. As this author has suggested above, the sermon is how the worshipping community's worldview, hermeneutic, and didoiesis are presented. Yet, Langley's homiletic method serves the hermeneutic principles. Again, this is not necessarily a bad practice. As a certain amount of hermeneutics are "presupposed truths", the homilist should consider whether or not her sermon is in service to the gospel or the hermeneutic.[26]

Wilson fares better. As the primary voice in homiletics, Wilson provides perhaps the most obvious homiletical contribution from the text. His homiletical rationales for the Law-Gospel hermeneutic are as follows (Gibson and Kim 2018, pp. 126–32):

1. The sermon is in two parts, starting from the law and ending in the gospel.
2. A gospel hermeneutic: the gospel is better than the law.
3. Identifying God's action for the theme sentence.

However, Wilson's application rationale provides a better method for the reader than the other contributors (Gibson and Kim 2018, pp. 132–45):

1. Four pages: (Buttrick's moves is perhaps the best comparison) (1) trouble in the biblical text, (2) trouble in the world, (3) grace in the biblical text, and (4) grace in our world.
2. Four sentences: each "page" is governed by a short sentence.
3. Sermon unity: this is ensured by having "one text, one theme from that text, one doctor in from that theme, one need from that theme, one image linked to the theme, and one mission that follows from them (Gibson and Kim 2018, p. 137)".
4. Proclamation: the main point, the climax of the sermon, "the" kerygmatic moment.

Wilson explores in the most simple of methods how to write a sermon with a clear beginning, middle, and end.

But what about the current state of Pentecostal homiletics? Here, Joseph K. Byrd provides the most telling state of Pentecostal homiletics in his essay *Pentecostal Homiletic: Convergence of History, Theology, and Worship*, where, in his conclusion, he offers what he believes to be the best homiletical option for Pentecostals (Green 2015). His suggestion is for Pentecostals to follow the methods of Eugene Lowry, specifically his works from the Homiletical plot (Green 2015, p. 286):

1. Upsetting the equilibrium
2. Analyzing the discrepancy
3. Disclosing the clue (the "aha" moment)
4. Experiencing the gospel
5. Anticipating the consequences

As the reader will notice, Lowry's use of post-modern language is not accidental: his method is directly against the deductive form in favor of an inductive form, which he calls "narrativity" (Green 2015, pp. 285–86). In this notion, Lowry suggests the minister should consider the "plot".

Does this truly matter, the language through which a homiletic is produced? It seems likely since homiletics is the means through which the worshipping community presents its worldview, hermeneutics, and didoiesis. By rejecting a deductive form or preaching, Byrd rightly asserts that,

> The goal is not to arrive at agreeing with a preacher's proposition, it is to hear the word of God in the aha moment in a way that resonates in the spirit of the listener and they a word of God to them in which they experienced the transforming truth of that Word... Adoption of an inductive model will correlate with a distinctive theology and history of the Pentecostal movement, contrasting the intellectual preaching of other Protestants with logical deductive reasoning. (Green 2015, p. 287)

Byrd's point is well taken. Yet, he perhaps "protests too much" against the idea of "argument". This author would not consider a persuasive narrative as "anti-argument". Indeed, both deductive and inductive forms seem to have their place within any "narrativity". After all, are not most stories vehicles for morals, perspectives, or ideas? Nevertheless, Byrd is not the only scholar to sketch the dissonance between deductive form and the Pentecostal's narrativity within her sermon.

## 5. Towards a Pentecostal Homiletic

### 5.1. Johns' Re-Enchantment

Cheryl Bridges John's most recent text *Re-Enchanting the Text: Discovering the Bible as Sacred, Dangerous, and Mysterious*, is something of a "wake-up call" for Pentecostals regarding how scripture is considered (Johns 2023). For Johns, Pentecostals have found themselves in the worst of predicaments: not truly Pentecostal and with more gnostics accidentally disguised as reformed rationalists (Johns 2023, p. 63). For Johns, scripture needs to be read *enchantedly*, or rather, Pentecostals should reject the Enlightenment's rationalist invasion into Protestant Christianity by re-embracing the reality of the Bible's transcendent nature and return to considering the Bible to be *Spirit-Word* and *Revelation*, both divine and human: *an enchanted text*.

To clarify Johns' *enchanted* etymologically: coming from the Latin verb *cantare* (to sing) preceded by the presupposition *in-*(into), produces *incantare*, meaning "finding oneself in a song" (Johns 2023, p. 55). It is a beautiful word that encapsulates John's primary argument: creation, and humanity, are part of God's imaginative unified cosmology, *eucatastrophe,* or a "good catastrophe" (Johns 2023, pp. 60, 62, 77). Reality is both natural and supernatural, physical and metaphysical, but not in a dualistic sense. Rather, Johns emphasizes that scripture is *magical*, although *mystical* is perhaps the better term. By *mystical*, it should be

understood that Pentecostalism recognizes that the divine and material affect the other, intertwined. Interdependent or codependent ideology misses the point. This is simply *how it is*: enchanted.

So, consider the worldview, hermeneutic, and didoiesis of Pentecostalism. These do not simply *express* the spirituality of Pentecostalism; they are *expressions* of the spirituality of Pentecostalism, as Pentecostalism emphasizes an enchanted view of not only Scripture but of human persons. Johns says it best:

> As spirit-word, scripture uniquely brings together the material with the divine. Humanity, likewise, is a unique synthesis of spirit and flesh. Persons do not exist as spirits who reside in the flesh, waiting for the great escape into the "real world. "This dualism inherited from the ancient Greeks, continues to plague Western Christianity. Rather, persons are a synergy of the material and the spiritual, prefiguring the resurrected life... if we envision human beings as embodied, desiring creatures, then it is easy to see how modern approaches to the Bible fail to engage us. Re-enchanting the natural world involves reuniting the material and spiritual dimensions of the cosmos. Re-enchanting Christianity involves sacramentally embodying grace within the life of the church. Re-enchanting the Bible involves bringing the Bible's material existence into the life of God's economy. Re-enchanting the people of God calls for bringing spirit and flesh together into a dynamic holism. The body, making visible the invisible mysteries of God's nature and redemption, serves as a means of revelation. Before scripture was written, it was spoken.... The written word and the spoken or living word or not two words. Rather they are two forms of the same word of God.... As embodied, sacramental beings, humans are made not only to enter the sacred space of scripture; we are also made to boldly inhabit the revelatory book of nature...human persons are made to live in an enchanted world. (Johns 2023, pp. 163–66, 168, 171)

In other words, God created creation and creatures to be sacramental Möbius strips, interwoven of both natural and supernatural (Johns 2023, p. 5).

This leads neatly to the primary question the Pentecostal is forced to consider: how does she write her sermon for/from/of such a worshiping community?

### 5.2. A Re-Enchanted Methodology

For the Pentecostal, homiletics is both discipline and discernment, scholastic and spiritual, method and mystery. As mentioned above, Pentecostals have a clear preference for principles above method; however, homiletics appears to be a potential synthesis for both principles and methodology; indeed, methods flow from principles.

Now, gleaming from Smith's worldview aspects, Green's preaching principles, Barth's sermon criteria, and Johns' (re)enchanted scriptures, with consideration to Buttrick's moves and structures, what are the governing principles to the Pentecostal preacher in her homiletical discipline? This author suggests the following acknowledgements[27]:

1.  Acknowledgment of God's substance within the text
2.  Acknowledgment of the Son's imaged divinity and humanity within the text
3.  Acknowledgment of the Spirit's activity within the hermeneut and hermeneutic
4.  Acknowledgment of the authority and enchantedness of the text
5.  Acknowledgment of the worldview, hermeneutic, and didoiesis of the worshipping community

The reader will surely see why "acknowledgment" begins each principle. Quite simply, the mental, physical, and spiritual prerequisites for sermon writing have to originate within the preacher herself. These acknowledgments appear to be "properly basic" in the vein of Plantinga's reformed epistemology, not necessarily needing any justification other than themselves. It seems obvious that each of these principles orients the preacher to a relational-reverence view of God, a high view of scripture, a humble view of ministry,

and a communal view of the sermon. As Smith, Green, and Johns' principles feed these acknowledgments, God, word, and community are elevated above the "content" of a sermon towards the "substance" of the sermon. In this, the intentions of the preacher might become the perceptions of the worshipping community. With these acknowledgments in mind, the preacher may now move to the actual homiletic method. To note, these acknowledgments also serve as the "narrativity" for the homilist as she interprets the text while interpreting herself and her community.

As explored previously, a methodology must be established in the presence of activity, especially in the hopes of accomplishing the predetermined task: writing the sermon. The following re-enchanted method proposes an 11-step process that emphasizes the importance of the minister's communion with God, the minister's responsibility to the text, personal dedication to the vocation, personal (honest) assessment in revisions, and personal awareness of her place within the worshipping community. These 11 steps also emphasize the active role of the Holy Spirit and the minister's role in discernment. This further communicates John's image of the homilist: a Möbius strip of natural and spiritual realities. This author suggests the following:

First invocation of the Holy Spirit: The homilist recognizes/invites the Holy Spirit's participation in the sermon preparation. This occurs through prayer, fasting, meditation, or any other discipline that intentionally focuses the homilist's heart, soul, and mind towards hearing God's voice through God's word and words.

1. Reading the text[28]: the homilist reads and is read by the text.
2. Re-reading the text with immediate observations: the homilist reads the text again, making general, intentional observations and notes about the text including authorship, themes, literary homilist, and references to other passages or texts.
3. Historical study: the homilist researches and considers the scope of the Church's interpretation of the text, both generally and in her worshipping community specifically.[29] The following hermeneutical methods are encouraged:

    a. The "four senses" of scripture: Scripture is read between and among its potential literal, moral, allegorical, and anagogical (mystical) senses (Levy 2018).[30]
    b. Archer's method: scripture is read between and among scripture, Spirit, and community.
    c. As a note, this is where the worldview, hermeneutic, and didoiesis inform the homilist's direction and sermon sundance unique to her worshipping community.

5. Second invocation of the Holy Spirit: the homilist seeks the guidance of the Spirit in discerning the appropriate delivery homiletical (preaching) method.

    a. Tone: the homilist considers her presentational attitude (e.g., encouraging, comforting, disciplinary, firm, etc.).
    b. Illumination: the homilist reminds herself to keep her own heart, soul, and mind to the activity of the Spirit for the message for the worshipping community for that time.

6. Writing the sermon: the homilist considers the moves and structure within the sermon and its place in the service, including the opening and closing sections of both sermon and service.

    a. Textual emphasis: the homilist considers her presentational method (e.g., keyword, analytical, comparative, etc.) (Hamilton 1992).
    b. The Wesleyan Quadrilateral: the homilist considers scripture, tradition (hers or others), reason, and experience (hers or others) in discerning the appropriate application of the enchanted truths with scripture for the worshipping community.

7. Editing: the homilist reviews her sermon, considering potential changes to apply to order, substance, or length. Additional considerations include: "altar time",[31] musical accompaniment, "prayer team" participation, etc.[32]

8.  Practice: the homilist practices her sermon, "feeling out" the moves.
9.  Final invocation of the spirit: the homilist thanks God for his word and his ministry through her within the worshipping community.
10. Preaching the sermon: the homilist, both humbly and boldly, preaches the sermon, continually discerning the Spirit during the service, the kerygmatic event.
11. Reflection: the homilist, following the service, considers the worshipping community's response and her own heart, mind, and soul, discerning the Spirit; reviews; and reflects. Additionally, the homilist should also listen to the worshipping community as they speak to her about the sermon, hearing their own thoughts and perceptions.

It should be observed that the homilist considers her worshipping community's identity as both being made by and as making the sermon, as for their perception of God, their neighbor, and themselves.

*5.3. Defending the Method*

It seems good to this author to provide supplementary explanations for the above 11-step process. While the 11 steps appear simple, or perhaps even ubiquitous to the Pentecostal, there are defining features that distinguish this method from others.

First, regarding the three invocations of the Spirit. These three moments serve as a type of discipline within a discipline. While it is true that the homilist is expected to "have the mind of Christ", she should also be an "imitator of Christ". In other words, she should be aware of how Jesus prepared for his ministry: he set out to pray[33]. The presentation of Christ and the Spirit cannot be overstated. Christ does not begin his earthly ministry until the Spirit descends upon him; it is the movement of the Spirit that drives him to the wilderness; it is by the power of the Spirit that he performs miracles and exorcisms; it is by (one could assume) the inspiration of the Spirit that he preached and taught. The Epistles further this notion of Christ-likeness commanding the homilist to "walk in the Spirit". For the Pentecostal, these innovations, to echo Brueggemann, re-orient her back towards God and his ministry of which she finds herself apart.

Second, regarding repeated engagement with the text(s) before the homilist begins writing her sermon: In the most mystical of senses, the Bible, as the reader has already seen, is God's word for his people. Again, returning to Buttrick does the homilist good: consideration of herself is essential in her sermon preparation. By returning to the text, the homilist consistently elevates the text above herself, stripping the reader of any biases, assumptions, etc., that could negatively influence the sermon for the worshipping community. This is why a historical study becomes vital to the homilist's discipline: not that the culture of the texts determines their meaning alone, but rather the culture of the text informs their meaning. Application of the text is not necessarily contingent upon the text's author, audience, or date of writing.

Third, regarding practice and reflection, both seem to this author to address the performative aspects of public communication. Here, this author wrestles to distinguish between presentation and performance. On the one hand, the sermon is a presentation, either deductive or inductive, and the homilist is making a presentation of a kind in which "acting" is not a consideration; however, on the other hand, the homilist is "playing", that is, "putting on a show" of sorts. Performance is not to be confused with pretense (or even as low as manipulation). However, if Byrd is to be understood, the homilist is telling a story, introducing the worshipping community and its neighbors back to God by way of encounter. Practice allows for familiarity, a comfortability even when the message may be uncomfortable. Reflection is similar: it is at once both a critique of the presentation/performance as well as listening to the worshipping community and God.

The five acknowledgments and the 11 steps also appear to fit with Smith's prescribed Pentecostal worldview and Green's preaching principles:

12. Radical openness to God *towards* theotic encounter: Acknowledgment 1 and Invocation Steps (1, 5, 9).

13.  "Enchanted" theology of creation and culture *towards* event of/occasion for the Spirit *and* Holy Spirit and (un)holy spiritedness: Acknowledgment 2 and Steps 2 and 3.

14.  Non-dualistic affirmation of embodiment and materiality *towards* overwhelmed, overwhelming askesis: Acknowledgment 3 and Steps 4, 6, 7, and 8.

15.  Affective, narrative epistemology *towards* protest for/against God: Acknowledgment 4 and Steps 10 and 11.

16.  Eschatological orientation to mission and justice *towards* kenosis and diakonia: Acknowledgment 5 and Steps 10 and 11.

## 6. Conclusions

As enchanted believers, the Pentecostal worshipping community should continually elevate scripture as both God's voice and God's words/word, authoritative and relational. The Pentecostal homilist precariously represents both God and the worshipping community back to themselves, recognizing that any distortion of the image falls upon the community to rectify. The homilist's task is both a wounding and healing of herself, her community, and God; it is a convicting and comforting activity, and though the homilist is responsible for her role within the kerygmatic event, she is not responsible for the community's reception or rejection of the word. There she stands; she can do no other.

**Funding:** This research paper received no funding.

**Informed Consent Statement:** Not applicable.

**Data Availability Statement:** Data is contained within the article.

**Conflicts of Interest:** The author declares no conflict of interest.

## Notes

[1]  The use of Church rather than church is intentional. By Church, I am referring to the universal worshipping community of those who "confess Jesus Chris, the Son of God, is Lord, believing that God rose him from the dead". Distinctions within the Church will be identified as needed.

[2]  Sproul explores this idea in his book Everyone's a Theologian: An Introduction to Systematic Theology emphasizing that as one engages with the character, activity, and being of God from Scripture, then one is theologizing.

[3]  Duo-personhood and tri-personhood are not the topics of this paper. This author understands personhood to be composed of both body and soul/spirit and will continue to refer to personhood as such since neither duo- no tri-personhood have any bearing on this paper's thesis.

[4]  Romans 1: 18–32.

[5]  Original word by author combining διδαχή (teaching, doctrine, what is taught) with ποίησις (a doing, making, performance; from ποιέω, meaning (a) I make, manufacture, construct, (b) I do, act, cause).

[6]  This author makes a distinction between dogma as the essential tenets of the Church/Christianity (non-negotiable truths) and doctrine as "in-house" disputes that are proven to be non-heretical and non-essential (negotiable stances).

[7]  Due to the broad and often convoluted history, theologies, and practices associated with and identified as pentecostal, this author has adopted what James K.A. Smith in Thinking in Tongues, referencing Steven Land's Pentecostal Spirituality suggests: pentecostalism, since it was not single-person led movement nor limited to a set of doctrines, should be better understood as a worldview. This explains the author's use of pentecostal (lower case "p") rather than Pentecostal (upper case "P").

[8]  Smith's distinction between -ity and -ism is immensely helpful in this discussion. -Ity suggests merely a quality, state or degree, whereas -ism suggests action, principles, doctrine, and even devotion.

[9]  Félix-Jäger and Shin emphasize that since pentecostalism is a renewal movement, it should be differentiated from other Christian worldviews. This seems to align with Smith's 5 key aspects of a pentecostal worldview and Land's pentecostal spirituality. To be clear, Félix-Jäger and Shin do not advocate for a pentecostal worldview in name, though it is near impossible to separate their work from a pentecostal worldview.

[10]  Ibid. xi. Félix-Jäger and Shin have wisely not defined "renewal worldview" as a "pentecostal worldview", but rather a potential rebranding of the framework. In doing so, Félix-Jäger and Shin present their "renewal worldview" as usable to all who are willing to adhere to it.

[11]  To be clear, there is no definitive "Christian worldview". Rather, "Christian worldview" simply means that Christianity has distinct and, perhaps, different answers to the fundamental questions that produce a framework through which a person lives and interprets that life.

12     Whereas reforming implies making a change to improve a thing, renewal implies either resuming a thing interrupted or changing a thing into something new or different. While both definitions provide valuable insight into the renewal worldview mechanics, they also provide a practical and theological platform.

13     Consider various movements within the Church: the monastic movement as a respondent to Constantine's Edict of Millan, the split of the Catholic Church from the Orthodox Church due to the Filioque Controversy, the Protestant Reformation, and so on. The fundamental reactions within the movements produced distinct worldviews within worldview, processes, and pedagogies.

14     Service here includes all liturgical, sacramental, and evangelical expressions and orderings of sabbath observations and rituals. While it is mainly within Protestantism that the sermon is central to the Service, all Services appear to contain preaching of some sort or another.

15     To be clear, there is certainly liturgy in both Catholic and Orthodox churches though Protestantism does not have a general "order of service".

16     To be clear, Green does not imply or suggest this explicitly. Rather, this is this author's application of his ideas.

17     Perhaps C.S. Lewis' analogy of man being amphibic (both land and water dwelling -> man being both physical and spiritual) provides a helpful visual.

18     This author uses the term performing in all seriousness of the word; the preacher is performing that which was prepared and practiced. Performing does suggest pretending or acting as in a play but as one would perform a speech or read a prayer from a text.

19     Barth is referenced no less than twenty times in Towards a Pentecostal Theology of Preaching.

20     It should be noted that Barth was adamantly against apologetics, which is perhaps why many pentecostals tend to cautiously accept his views, despite their reformed slant.

21     Buttrick acknowledges that most would call these "points" of sermons, however, this language misses the "point" itself.

22     For Buttrick, moves truly is the "before the preaching" work of homiletics.

23     The author is loathed to consider the substance of sermons as content.

24     Gibson: of the 163 pages, only 40 odd pages directly reference homiletical methods, once again adopting principle over method as the preferred discussion.

25     This is no attack against Chappell or Kuruvilla: hermeneutics merely appears to be their primary concern. Their "Homiletical and Application Rationale" fails to impress.

26     This author recognizes that each minister will have to maintain an honest awareness of whether or not their denominational doctrines are in service to the Gospel or if the Gospel is in service to their denominational doctrines.

27     By "acknowledgment," this author is using its definition to "accept or admit the existence or truth of" and "to recognize the fact or importance or quality of" with special emphasis on the latter usage. Acknowledgment, therefore, becomes uniquely attributed to the renewal worldview of pentecostals as this notion aligns itself with presuppositions or even pre-critical consideration.

28     This author recognizes the various traditions involved with text selection for preaching. However, regardless of the text selection process, this step remains intact.

29     Example: homilist would consider a Lutheran reading of the text while not ignoring her own

30     Levy provides perhaps the best explanation and exploration of this method.

31     A traditionally pentecostal conclusion to a message allows the worshiping community to immediately respond in earnest to the message.

32     These considerations are not "dictated". The homilist is to be sensitive to the Spirit in all stages of the sermon: preparation, presentation, and post-service response and reflection.

33     This author will refrain from exploring the discipline of prayer but encourages readers to read Prayer by Richard J. Foster.

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
