# Peer review of "Towards a Pentecostal Homiletic: A Re-Enchanted Methodology"

_religions, doi:10.3390/rel15010045_

Round 1

Reviewer 1 Report

Comments and Suggestions for Authors

Overall, I found the essay intriguing in terms of both concept and method. I think the stereotype of Pentecostal preaching is that it is more ambiguous when it comes to method of preparation or homiletical structure. This essay demonstrates that this is not (or should not be) the case in contemporary ecclesiology.

As a missional endeavor, Pentecostalism, as I understand it, focuses on preaching the gospel clearly and convincingly -- which the method articulated in this essay argues for. Its analysis of both Pentacostal and mainline Christian literature (especially the analysis of Barth) is compelling and demonstrates an astute reading.

Of special note was the invention of the author's phrase didoiesis (p. 2, l. 63). I found this concept extremely intriguing, especially as one who takes a more rhetorical and pedagogical approach to preaching. I would encourage the author to continue developing this concept.

Aside from some general editing (i.e., misspelling of instead on p. 4, l. 110; identifying clearly H2 in diagram on p. 4), my main concern is the use of male-dominated language in the opening section (1.1). I think this language could largely be revised as "human" or "humanity" to demonstrate inclusion without losing its epistemological significance.

Overall, this was a thoughtful and compelling essay that, I think, speaks to a missing and needed area of homiletic and ecclesial thought!

Comments on the Quality of English Language

Overall, aside from some minor revisions, the quality of English was strong.

Author Response

Hello Reviewer 1,

Thank you for your time and energy in reading my paper. I appreciate and have considered all your suggestions. 

The changes you suggested have been applied and highlighted in BLUE. Please the attached pdf. 

Reviewer 2 Report

Comments and Suggestions for Authors

The presented article introduces a significant perspective in the field of homiletics, particularly within the context of Pentecostal practical theology. It is commendable that the author seeks to address this gap in research, given the current lack of contributions on this subject.

Undoubtedly, the article's strengths lie in its innovative perspective and potential enrichment of the homiletic discourse. Therefore, I unequivocally recommend publication after a thorough revision, as the contribution can substantially impact homiletic research.

However, a central weakness of the article resides in its reader guidance, which needs improvement. To address this, I provide specific examples: the abstract could benefit from a more precise presentation of the main question, objectives, and results, providing readers with orientation. Additionally, enhancing transitions between chapters would be highly beneficial. Readers need to understand why a particular author is being discussed and why that individual is crucial to the article's main theme or central question. A detailed explanation of the criteria for selecting these authors would enhance the hermeneutics of the article, making the relevance of these sources more explicit to readers. Both Barth and Buttrick are no longer central to current homiletic discourses. A more precise rationale for including these authors in the context of Pentecostal homiletics would improve the hermeneutic disclosure of the program.

Regarding the methodology (Section 5.2), the author should elaborate on how the 11 points of Pentecostal homiletics were developed, how they relate to the discussed authors and the overall context of the article. In essence, all these critiques are tied to reader guidance. I believe the effort to enhance this guidance is manageable. It involves making transitions clearer and occasionally inserting explanatory sentences to help readers understand the author's approach better.

Two Additional Points for Consideration:

  1. In Chapter 1, a predominantly masculine language was employed. However, in the subsequent chapters, this is no longer the case (which, in my opinion, is positive). Nevertheless, this linguistic shift regarding gender themes between Chapter 1 and the rest creates a noticeable discontinuity. I would recommend adopting a more inclusive language for Chapter 1 as well to maintain consistency throughout the article.

  2. Another point to address is the absence of engagement with current homiletic concepts. In my view, exploring contemporary homiletic ideas could provide valuable insights and suggestions for a Pentecostal homiletic, as presented by the author. The article would improve quality by positioning itself more distinctly within ongoing homiletic discussions. Integrating contemporary debates would further enhance the article's overall quality.

In conclusion, I strongly encourage the author to revise and publish this article. Through careful revision, a clearer and more compelling presentation of the Pentecostal perspective depicted here can be achieved. The work holds the potential to make a significant contribution to homiletic research, and meticulous revision could further strengthen its impact.

Author Response

Hello Reviewer 2,

Thank you for your time and energy in reading my paper. I appreciate and have considered all your suggestions. 

The changes you suggested have been applied. 

The changes have been highlighted in BLUE. See attached pdf. 
